# Effects of Manufacturing Variation in Electronic Cigarette Coil Resistance and Initial Pod Mass on Coil Lifetime and Aerosol Generation

**DOI:** 10.3390/ijerph18084380

**Published:** 2021-04-20

**Authors:** Qutaiba M. Saleh, Edward C. Hensel, Nathan C. Eddingsaas, Risa J. Robinson

**Affiliations:** 1Department of Computer Engineering, Rochester Institute of Technology, Rochester, NY 14623, USA; qms7252@rit.edu; 2Department of Mechanical Engineering, Rochester Institute of Technology, Rochester, NY 14623, USA; rjreme@rit.edu; 3School of Chemistry and Materials Science, Rochester Institute of Technology, Rochester, NY 14623, USA; ncesch@rit.edu

**Keywords:** aerosol generation, e‑cigarette, coil resistance, e-liquid, manufacturing variation

## Abstract

This work investigated the effects of manufacturing variations, including coil resistance and initial pod mass, on coil lifetime and aerosol generation of Vuse ALTO pods. Random samples of pods were used until failure (where e-liquid was consumed, and coil resistance increased to high value indicating a coil break). Initial coil resistance, initial pod mass, and e-liquid net mass ranged between 0.89 to 1.14 [Ω], 6.48 to 6.61 [g], and 1.88 to 2.00 [g] respectively. Coil lifetime was µ (mean) = 158, σ (standard deviation) = 21.5 puffs. Total mass of e-liquid consumed until coil failure was µ = 1.93, σ = 0.035 [g]. TPM yield per puff of all test pods for the first session (brand new pods) was µ = 0.0123, σ = 0.0003 [g]. Coil lifetime and TPM yield per puff were not correlated with either variation in initial coil resistance or variation in initial pod mass. The absence of e-liquid in the pod is an important factor in causing coil failure. Small bits of the degraded coil could be potentially introduced to the aerosol. This work suggests that further work is required to investigate the effect of e-liquid composition on coil lifetime and TPM yield per puff.

## 1. Introduction

The aerosol generated by an Electronic Nicotine Delivery System (ENDS) or electronic cigarette depends on the electrical characteristics of the heating element (hereinafter referred to as the coil [1,2,3,4,5]), the characteristics of e-liquid [6,7,8,9], the ENDS Power Control Unit (PCU) and battery [10,11,12,13], user behavior such as puff flowrate and puff duration [10,14,15,16], and device design (e.g., internal geometry, flow path design, wick and coil design and locations) [10,14]. It has been widely reported that altering one or more of these factors could change the aerosol emissions from the ENDS such as the total particulate matter (TPM) yield, hazardous and potentially hazardous constituents (HPHC) of the aerosol, and consequently the health effects. In particular, changing coil resistance, which results in altered power consumption, has been associated with changes in some carbonyls and reactive oxygen species (ROS) [1], changes in concentration of selected aldehydes [1,2,3], changes in nicotine delivery, and changes in puff topography and e-liquid consumption [4,5].

The variation in coil resistance has the potential to change the power consumed in the coil and thus the amount of heat generated. Consequently, it could change the performance of the ENDS, which in turn leads to changes in aerosol emission and presence of the HPHCs. Less attention has been given to the manufacturing variability of coil resistance and the relative impact on variations in aerosol yield. Our previous work [17] introduced a robust method to measure coil resistance of e-cigarettes and documented manufacturing variation in coil resistance of two popular pod-style ENDS: Vuse ALTO and JUUL. Pod units included in the test showed variation in coil resistance of ~30% and ~7.4% for ALTO and JUUL, respectively. Several studies have reported manufacturing variations in e-liquid characteristics [18,19,20,21,22]. In preliminary work for this study, we observed manufacturing variations in the gross mass of Vuse ALTO pods, thought to be dominated by the amount of e-liquid in the pod rather than by differences in the container mass.

More study is needed to better understand manufacturing variation and the impact of this variation on device performance. This study leads to better understanding of the impact of manufacturing variation on coil lifetime and aerosol emissions. Therefore, this study begins to lay the groundwork for regulations requiring manufacturers to report variations in tobacco product components as part of the premarket approval process.

### Study Objectives

This study focused on investigating the effects of manufacturing variations on e-cigarette performance; specifically: (1) the effects of the variation in initial coil resistance on coil lifetime and TPM yield per puff, and (2) the effects of the variation in initial pod mass on coil lifetime and TPM yield per puff.

## 2. Materials and Methods 

In order to investigate the effects of initial coil resistance and initial pod mass on device performance, this study measured coil lifetime and total particulate matter (TPM) yield per puff. Coil lifetime was measured as the number of repeated puffs delivered from a brand-new pod until the coil broke, without refilling the pod, as indicated by a sharp increase in coil resistance. Coil breakage could be a result of coil aging, excessive usage, or a result of energizing the coil in the absence of e-liquid. The excessive heat generated by the coil could lead to melting and breaking the coil. All of these failure mechanisms would be reflected in such a sharp increase in coil resistance. Therefore, coil resistance was deemed to be a good indicator of coil breakage.

### 2.1. Test Specimens

The experiments were conducted on a commercially available pod style ENDS, Vuse ALTO [23] pods *N* = 15 which is one of the most popular e-cigarettes among teenagers [24,25]. The manufacturer reported that these pods are filled with 1.8 [mL] of e-liquid. The pods used in this study filled with nicotine flavor e-liquid manufacturer labeled 5% nicotine concentration. They were purchased from local retail shops and national online vendors.

### 2.2. Aerosol Generation and Collection

The previously validated Programmable Emissions System™ (PES™-1) was used to activate and run the ENDS under test to generate and collect aerosols as described in [26]. The system can be configured to perform puffing profiles based on a wide range of puff flowrates, puff durations, and inter-puff intervals. It uses a vacuum tank (5.0 [L] with pressure as low as −60 [kPA]), a proportional valve (KPIH-VP-20-156-25, Kelly Pneumatic Inc., Newport Beach, CA, USA with 10 [m s] response time) and a gas flow meter (M-50SLPM-D-30PSIA/5M, Alicat Scientific, Inc. Tucson, AZ, USA) connected in series to generate the required flowrates. The flow meter and proportioning valve are digitally monitored and controlled to implement the desired puffing profile. Several particulate phase collection modules can be used with this system. The current study used Cambridge style single stage filter pads.

The TPM collected during each trial was measured by differencing the mass of the filter pad before and after the trial. A Mettler AE240 Analytical Balance gravimeter was used. The Mettler balance provided a protected weighing space with accuracy of ±0.0002 [g] (0.2 [mg]) [27]. The same gravimeter was also used to measure pod mass before and after each trial.

Figure 1 shows the entrance region of the experimental setup, which includes the inlet of the PES-1, filter pad holder, short connecting tube, and the ENDS under test. The PES-1 was set up with an angle of 30° to mimic the declination angle of the ENDS while being puffed, determined in a previous Master’s thesis, which analyzed data from YouTube videos of e-cigarette users while vaping their personal ENDS in their natural environment [28]. The mouthpiece of the Vuse ALTO ENDS was connected to the inlet of the filter pad holder through a short connecting tube. The tube was fixed to the mouthpiece with Bemis^TM^ Parafilm^TM^ M Laboratory Wrapping Film (not illustrated in the picture for visibility).

### 2.3. Coil Resistance Testing Apparatus

The test fixture presented in [17,29] was used to measure coil resistance, built by repurposing the housing of the PCU (power control unit) of the targeted ENDS, mimicking the geometrical and electrical conditions of the original ENDS. The test fixture provides measurement of the effective coil resistance which accurately represents the resistance seen by the PCU during operation. The effective coil resistance is the summation of the resistances of the connectors from the PCU to the pod, the internal connection pins in the pod, and the heating coil. The fixture utilizes a four-wire resistance measurement configuration with customized test leads. A detailed step-by-step protocol for building this fixture has been published on protocols.io [29]. The fixture was used with 34465A KEYSIGHT™ Digital Multimeter [30].

The coil resistance test fixture [17,29] was held vertically using a tabletop vise to ensure consistency in the measurement and minimize error resulting from motion. The test fixture was connected to the digital multimeter and communicated with the PES-1 personal computer via USB serial connection. When the pod was inserted in the test fixture and the resistance measurement was ready to be made, a button in the PES-1 software could be clicked to make the coil resistance reading and record the results in the dataset.

### 2.4. Data Acquisition 

Several types of data were collected while conducting the experiment, including the measured puffing profile (flowrate), labeling data about the ENDS and pod under test, filter pad mass, pod mass and coil resistance. The measured flowrate was automatically collected by the PES-1 controller software, which was saved as a comma-separated values file at the end of the session for later usage. The PES-1 controller software also provided the means to enter the other types of the data. The labeling data of the ENDS and pod under test were scanned by a barcode scanner before the session, the filter pad mass and pod mass values were manually measured using the gravimeter and were manually entered to the PES-1 controller software before and after the session (i.e., after 20, 10 or 5 puffs). Coil resistance was read by the PES-1 controller software when the pod was inserted in the test fixture before and after each session. The step by step testing procedure employed in this study has been published [31] to foster the reproducibility of this work.

### 2.5. Puffing Profile

The lifetime testing puffing profile used here included emissions testing sessions with uniform rectangular shape puffs whose puff flowrate was 18.33 [mL/s], puff duration was 5.5 [s], and puff interval was 11 [s]. The number of puffs per session was 20 puffs for the earlier portion of coil lifetime and was reduced to 10 or 5 puffs per session as each coil lifetime test progressed. This puff profile was designed to accelerate lifetime testing by providing long puff duration and short puff interval in order to shorten the time required to fully consume the pod and achieve coil failure. Such technique (accelerated lifetime testing) has been used in quality assurance testing standards of many common products [32,33]. The profile was also carefully designed to consider the parameter margins suggested by the manufacturer, in order to avoid interfering with the results of the experiment while trying to comply with some aspects of the Cooperation Centre for Scientific Research Relative to Tobacco (CORESTA) standard for e-cigarette aerosol generation and collection [34]. The flowrate was chosen based on the results of a preliminary experiment done in our lab which showed that Vuse ALTO ENDS was consistently activated at a flowrate of ≥15 [mL/s]. While the puff flowrate complies with CORESTA standard, it was also intentionally selected to be low in order increase the aerosol generation efficiency. The puff duration of 5.5 [s] was chosen to fully exercise the five seconds specified by the manufacturer before the ENDS automatically stops puffing [35], while the CORESTA standard specifies a puff duration of 3 ± 0.1 [s]. The puff interval (11 [s]) was shorter than the CORESTA interval (27 [s]). This 11 [s] puff interval is longer than the time specified by the manufacturer to cool the device after it has been used for ≥5 [s] puff duration. Complying with the CORESTA flow rate was chosen to make it easier for other researchers to compare our results. We are, however, not suggesting either the CORESTA profile or the accelerated lifetime puffing profile used herein accurately represents human user behavior. This profile might not be suitable for other experiments that focus on different research objectives or test different devices.

## 3. Results

### 3.1. Illustration of Coil Lifetime

Coil lifetime was defined as a sharp increase in coil resistance wherein the coil melts or disconnects. Verification of a sharp increase in coil resistance as a measure of coil lifetime was demonstrated by dissecting and inspecting three Vuse ALTO pods with different levels of usage. Figure 2 shows pictures of Vuse ALTO coils with three different conditions: New coil, Pre-Failed and Failed. The ALTO coils are ‘S’ shaped metal strips on a porous ceramic wick substrate. Two metal connectors, evident as the circular area on the left and right side of each image, are mounted to the terminals of the coil and connect the coil to the power control unit. The new coil had never been used and exhibited no sign of wear. The pre-failed coil had been used until the pod appeared visually empty of e-liquid. It was, however, a working coil with a functional resistance value. The pre-failed coil exhibited signs of erosion and oxidation especially in the lower portion of the ‘S’. The failed coil had been used until its resistance value increased to ~400 [kΩ] indicating coil failure. The failed coil exhibited severe signs of wear and a complete physical break in the metal ‘S’ coil can be easily seen below and to the right side of the left terminal.

### 3.2. Impact of Initial Coil Resistance on Coil Lifetime and TPM Yield

The first objective was to investigate the effects of variation in initial coil resistance (prior to first puff) on coil lifetime (measured as number of puffs until coil failure) and TPM yield per puff. Figure 3 shows coil resistance values for each session as a function of cumulative puff count. The data points are presented as a scatter plot of coil resistance vs. cumulative puff count, overlaid by a boxplot of the same data in an effort to understand changes in variation over the course of coil life. The first six emissions sessions for all pod specimens consisted of 20 puffs per session. Thereafter, the operator reduced the count from 20 to 10 to 5 puffs per session as the e-liquid remaining in each pod decreased. The initial coil resistance of the pods ranged between 0.89 [Ω] and 1.14 [Ω] with sample mean (µ) = 1.02 [Ω] and standard deviation (σ) = 0.081 [Ω]. Coil resistance was relatively steady for the first 120 puffs (~ first 6 sessions). After 135 puffs, coil resistance values started to increase as some pods started to exhibit coil failure and the number of scatter points decreases as a function of cumulative puff count. After 165 puffs, only 3 coils remained operable, while after 190 puffs only one coil remained in operation. Coil lifetime varied between pods from 135 puffs to 215 puffs with µ = 158 puffs and σ = 21.5 puffs.

We generated a scatter plot (not shown) of this data and conducted linear regression analysis to investigate a possible association between initial coil resistance and coil lifetime. There was insufficient evidence to support an association between coil lifetime and initial coil resistance (r = −0.07, *p* = 0.79). Next, we generated a scatter plot (not shown) to investigate a possible association between initial coil resistance and initial TPM yield per puff (first session). The initial TPM yield per puff ranged from 0.0118 [g] to 0.0129 [g] with µ = 0.0123 [g] and σ = 0.0003 [g]. We found no evidence to support this relation (r = −0.26, *p* = 0.35).

### 3.3. Impact of Initial Pod Mass on Coil Lifetime and Coil Resistance Variation

The second objective was to investigate the effects of variation in initial pod mass on coil lifetime and TPM yield per puff. Figure 4 shows coil resistance values vs. pod mass at each session starting from a brand-new full pod to coil failure point. The gross mass of the brand new pods (initial pod mass) ranged from 6.48 [g] to 6.61 [g] with µ = 6.54 [g] and σ = 0.0469 [g] while the tare mass of the pods after failure (end pod mass) ranged from 4.56 [g] to 4.67 [g] with µ = 4.61 [g] and σ = 0.0342 [g]. During the full exhaustive test, the net mass of the e-liquid consumed out of each pod ranged from 1.88 [g] to 2.00 [g] with µ = 1.93 [g] and σ = 0.035 [g]. The connecting lines between scatter points are used as a visual aid to show that coil resistance remains relatively steady for most of the session series, while e-liquid remained in the pods. However, coil resistance values initially decreased as the e-liquid level approached the wick and then sharply escalated, indicating coil failure, at which point the pods visually appeared completely empty. We generated scatter plots (not shown) of this data and conducted linear regression analysis to investigate a possible association between initial pod mass and coil lifetime. We found insufficient evidence to correlate coil lifetime with either initial pod mass (r = 0.03, *p* = 0.9) or the net mass of e-liquid consumed (r = −0.2, *p* = 0.45).

### 3.4. Impact of Initial Pod Mass on TPM Yield

Figure 5 shows the TPM yield per puff vs. pod mass at each session starting from a brand-new full pod to coil failure point. As expected, the TPM yield per puff approaches 0 when the e-liquid in the pod is fully or almost fully consumed. We generated scatter plots (not shown) to investigate a possible association between initial pod mass and initial TPM yield per puff. We found insufficient evidence to correlate TPM yield per puff with initial gross pod mass (r = −0.23, *p* = 0.41) or with net mass of e-liquid consumed (r = 0.08, *p* = 0.76).

## 4. Discussion

### 4.1. Why Did Initial Coil Resistance Not Affect TPM Yield?

The results presented herein demonstrated insufficient evidence to support correlations between initial coil resistance TPM yield per puff. In the simplest design of an ENDS, the PCU simply short-circuits the battery across the coil and there should be a correlation between the TPM yield per puff and the initial coil resistance when the ENDS battery is fully charged. There are at least two possible explanations for the lack of correlation. One explanation is that the PCU employs an algorithm which limits power dissipated in the coil, either through voltage control, current control, puff duration control, or duty cycle control. In this case, the temperature of the coil will be limited to the boiling point and further increase of the heating power should not affect the temperature as long as there is a liquid in contact with the coil. However, if the applied power increases, the additional power would increase TPM yield even in the absence of an increase in coil temperature. A second explanation is that the PCU employs an algorithm that limits the maximum coil temperature to prevent over-heating of the coil. In this case, we would not have observed the coil burn-out failure exhibited in Figure 3. Thus, we infer the ENDS PCU did exhibit some level of power control but did not exhibit an over-heating protection circuit.

This suggests that some ENDS PCU may employ algorithms to overcome variation in coil resistance or actively control the power. Such algorithms could use a closed-loop control system which dynamically measures coil resistance and adjusts the power supplied to the coil in real time in order to keep the heating energy within a limit, as has been previously disclosed in the patent literature [36,37], and research literature [38,39]. Thus, the temperature of the coil and heating chamber could be controlled in a cycle to keep aerosol emission steady. No articles have been presented in the literature which quantify the effectiveness of such PCU algorithms and the extent to which they can eliminate the effects of coil resistance variation on the performance of the ENDS. The results presented herein establish a firm premise for the study of PCU control algorithms. Such PCU algorithms offer potential for both positive and negative health effects, and thus are worthy of detailed investigation and possible regulatory action.

### 4.2. What Are the Mechanisms of Coil Failure?

The results demonstrate that a dramatic increase coil resistance is a sufficient indication of coil failure. At failure point, the coil melts or breaks leaving an open circuit between its terminals. This break is reflected as a sudden increase in measured coil resistance from order of ~1 [Ω] to order of ~400 [KΩ]. While our definition of coil failure suggests coil break, our measured coil resistance was not infinite. The residual resistance measured after the failure point could be related to the resistance of the wick and some drops of e-liquid that might still exist around the wick. We observed that the ENDS continued to operate during this pre-failure condition, when the coil was degraded (Figure 2, middle image) and loss of metal was observed. The results presented herein may explain the mechanisms underlying coil failure. When the coil is active, it generates heat that is transferred to the e-liquid causing it to vaporize. Concurrently, the e-liquid cools the coil as generated aerosol carries the heat away. This suggests that the presence of e-liquid around the coil contributes to limiting the coil temperature below its melting point, consistent with the observed steadiness in coil resistance values while there is e-liquid left in the pod (Figure 4). When insufficient e-liquid remains to fully submerge the coil and the wick, the heat generated by the coil remains in the coil and the wick causing the coil temperature to increase and thus the coil melts or breaks. A follow-up experiment could be conducted to confirm this explanation by demonstrating the coil lifetime can be extended indefinitely by refilling the e-liquid reservoir, even though the pods studied here are intended by the manufacturer to be disposable. Whether or not manufacturers are required to protect against product misuse, significant public health concerns may arise therefrom, and the proposed experiment may inform future research into lung injury and atypical health responses observed among ENDS users.

### 4.3. Potential Health Impact of Coil Failure and Product Misuse

Metal is likely ejected from the pod into the aerosol while the coil fails during the final puffing session and potentially much earlier. The coil shown in the middle picture of Figure 2 illustrates this degraded condition wherein chunks of the coil are gone while it continued to generated aerosol. The chunks of the degrading coil are most likely to be ejected with the aerosol and inhaled by the user. This observation is consistent with results of several articles which test the existence of metal in electronic cigarettes’ emission [40,41,42,43,44]. This is a potentially critical juncture, particularly in the instance where consumers misuse their product and refill the e-liquid in a pod which was nearly emptied on a prior use and whose coil had experienced degradation. When a compromised coil is subsequently heated, there may be increased risk of metal exposure even when the coil has not fully failed. Follow-up experiments to assess the emissions effects of e-liquid refilling could eventually establish a scientific foundation for regulations requiring labels warning against this type of misuse or product safety/inter-lock features preventing re-filling, or both.

### 4.4. Limitations and Future Work

Visual inspection of pods which were just taken out of the consumer over-packing (not exposed to light or air since packaged by the manufacturer) revealed variations in e-liquid color among pods in different blister-packs and between pods in the same blister-pack as shown in Figure 6. The pods were primitively classified as “light” or “dark” in color (prior to running experiments) by visually comparing the pods with each other. Out of 15 ALTO pods used in the experiment, 11 pods were classified as “light” while 4 were classified as “dark”. While analyzing the data, it was noticed that coil lifetime of pods with light color e-liquid appeared to be consistently shorter than that of dark color e-liquid as shown in Figure 3. In this figure, the pods classified as light color noted with ‘*’ while pods classified as dark color noted with ‘o’. The light color group was found to have coil lifetime in the range of 135 to 165 with µ = 149, σ = 10.7 puffs while the dark color group had coil lifetime in the range of 165 to 215 with µ = 185, σ = 22.7 puffs. An *a-posteriori* t-test between the two groups showed a difference of 36 puffs (*p* < 0.001), confirming an association between coil lifetime and e-liquid color. The TPM of the pods in the light e-liquid group and the dark e-liquid group behave differently while the e-liquid is being consumed as shown in Figure 5. The TPM yield per puff of the light e-liquid group, ‘*’, is relatively steady for most of the sessions and sharply decreases when the e-liquid is fully consumed or almost consumed. On the other hand, for the dark e-liquid pods, ’o’, the TPM yield per puff gradually (linearly) decreases while more e-liquid is being consumed until it sharply decreases just before the e-liquid is fully consumed. These observations were not further analyzed as they lack the appropriate chemical analysis, which explains the variations in e-liquid color. A study with the appropriate measurements is being undertaken to closely investigate this variation and its effects on ENDS performance.

## 5. Conclusions

We found insufficient evidence to correlate coil lifetime and TPM yield to either initial coil resistance or initial pod mass. The amount of e-liquid remaining in the pod appears to be the single most important factor in determining coil failure. A dramatic sharp increase in observed coil resistance is a robust method for quantifying coil lifetime. Further investigation is needed to assess the potential adverse health impacts of coil degradation during the final stage of coil lifetime. This work suggests that further work is required to investigate the effect of e-liquid composition on coil lifetime and TPM yield per puff.

## Figures and Tables

**Figure 1 ijerph-18-04380-f001:**
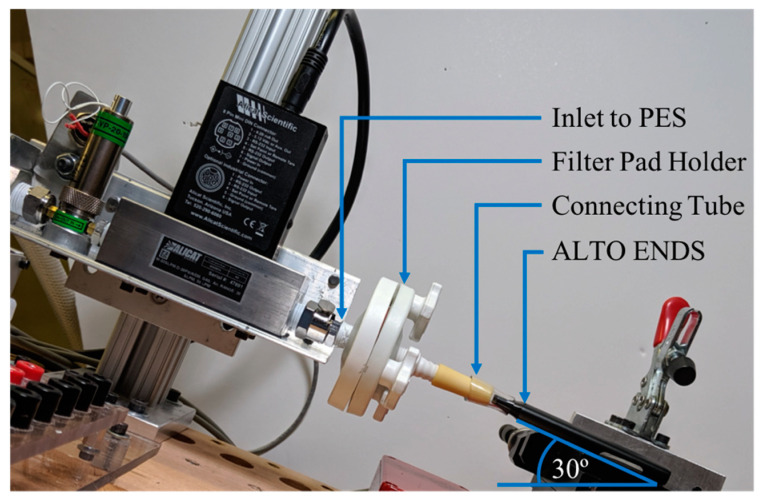
PES-1 setup with ALTO connected to it at 30° angle through a filter pad holder.

**Figure 2 ijerph-18-04380-f002:**
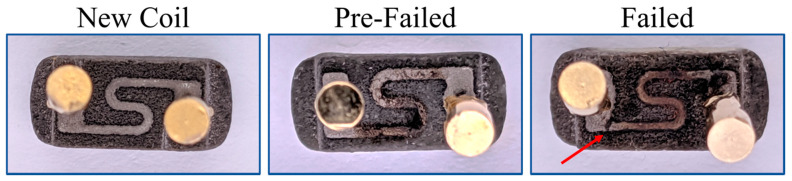
Pictures of ‘S’ shaped Vuse ALTO coils with three different conditions. ‘New Coil’ is a never used coil, ‘Pre-Failed’ is a coil which had been used until the pod appeared nearly empty of E-Liquid, and ‘Failed’ is a coil which had been used until its re resistance value increased to ~ 400 [KΩ].

**Figure 3 ijerph-18-04380-f003:**
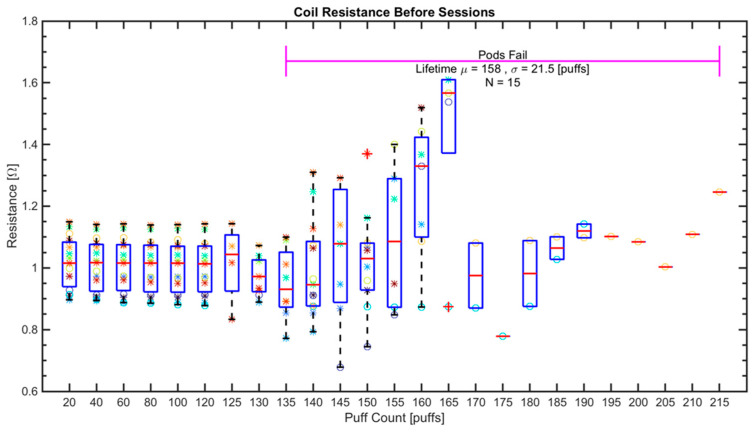
Coil resistance over time (no. puffs) starting with brand new full pod until failure for the *N* = 15 pods tested in this study. Data is represented as scatter plot where each pod is represented by a different marker color. Data is also represented with a box plot where the horizontal red marker indicates the group mean and a red plus marker indicates the corresponding data point is an outlier.

**Figure 4 ijerph-18-04380-f004:**
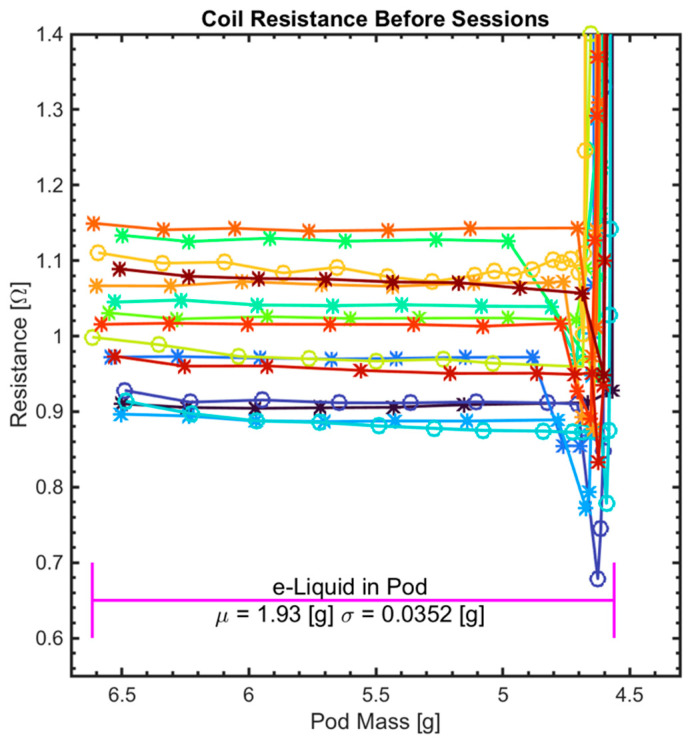
Coil resistance value vs. pod mass before each session starting from brand new full pod until failure where the pod visually looks empty for *N* = 15 pods tested in this study. Each pod is represented by a different marker color.

**Figure 5 ijerph-18-04380-f005:**
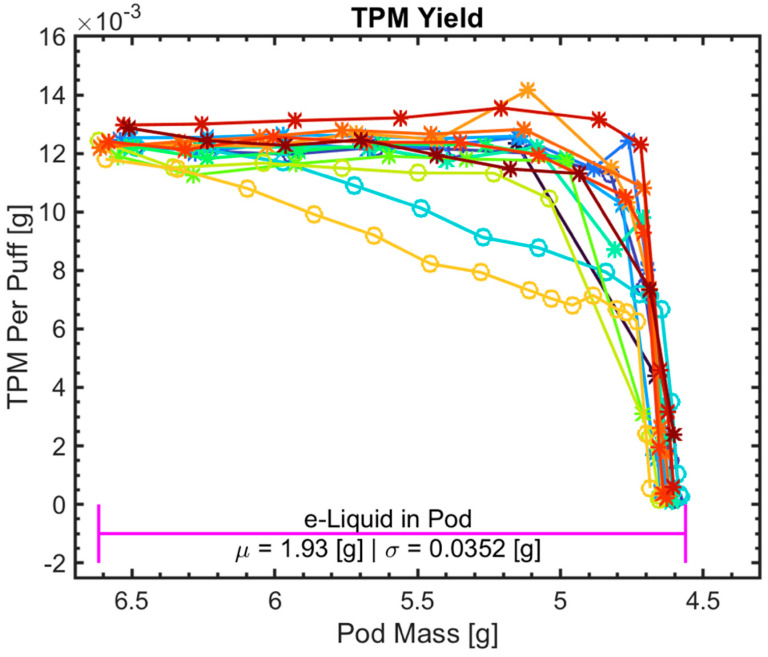
TPM yield per puff vs. pod mass for each session starting from brand new full pods until coil failure where the pod visually looks empty for *N* = 15 pods tested in this study. Each pod is represented by a different marker color.

**Figure 6 ijerph-18-04380-f006:**
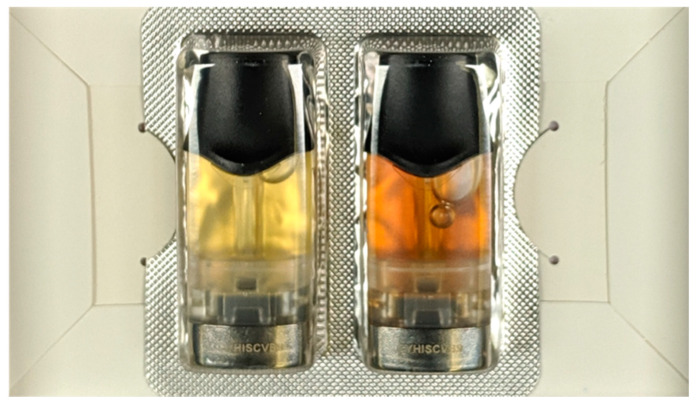
Nicotine flavor Vuse ALTO pods with 5.0% nicotine concentration. The two pods in the same pack have e-liquids with two different colors. The left pod was classified as light color while the right pod was classified as dark color.

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
