# Peer review of "Effects of Manufacturing Variation in Electronic Cigarette Coil Resistance and Initial Pod Mass on Coil Lifetime and Aerosol Generation"

_ijerph, 2021, doi:10.3390/ijerph18084380_

Round 1

Reviewer 1 Report

Comments to Authors:

This paper is focused on an interesting observation regarding the influence of e-liquid color on ENDS performance (although this was not the initial goal of the study). Authors tried to provide explanations of the observed effects but some of these explanations look not enough justified.

The weakest side of this study is an absence of the chemical analysis of the e-liquid. Authors indicated that this is a subject of undergoing study but as of now it seems like explanations presented by authors regarding the color effects that they observed are rather hypothetical.

In my personal opinion this paper would benefit if it will be published together with chemical analysis but it’s up to authors to decide.

More detailed comments please see below:

L32-35: The aerosol generated by an Electronic Nicotine Delivery System (ENDS) or electronic cigarette depends on the electrical characteristics of the heating coil [1-5], the characteristics of e-liquid [6-9], the ENDS Power Control Unit (PCU) and battery [10-13], and user behavior such as puff flowrate and puff duration [10,14-16].

Comment: although it might not be reflected in the current published literature yet but internal geometry of the ENDS (such as configuration of the heating zone and aerosol delivery channel) may also play an important role.

L184-185: The puff interval (11 [Sec]) was shorter than the CORESTA interval (27 [Sec]). Complying with the CORESTA flow…

Comment: This interpuff interval seems very short and may represent unrealistic ENDS usage conditions. If temperature of the coil did not have enough time to cool down it might affect coil life-time.

L363-365: The saturation temperature of nicotine is 247.0 ï‚°C [47] while the saturation temperature of nicotyrine is 281.0 ï‚°C [48]. Not only does this mean nicotyrine vaporizes at higher temperatures, it also indicates that e-liquids with higher NNR would have lower TPM yield compared to those with lower NNR, particularly if 366 the coil temperature is above 247 C and below 281 C.

Comment: I would assume it is more complicated. Since the amount of nicotine or nicotyrine is low (5%) in comparison with PG and GL mass it is unlikely that just a difference in saturation temperature itself would affect the TPM. The other thing that here we are dealing with multicomponent nucleation (phase transition from vapor to aerosol phase). From what is known about nucleation adding of another compound could seriously affect nucleation rate. So, theoretically it could happen that change of nicotine to nicotyrene in the PG/GL based solution will affect nucleation rate. But properly designed experiment should be conducted to confirm this assumption.

L369-374: And, as the effective saturation temperature increases, the yield per puff will decrease and the number of puffs required to empty the pod, expose the coil, and induce failure will increase. For this reason, and if all other factors are held constant, we should expect pods with dark color liquid to exhibit lower TPM and longer coil lifetime than pods with light color e-liquid, consistent with the results reported in our study.

Comment: I guess I don’t understand this logic. As the saturation temperature increases it should vaporize main components of e-liquid (PG and GL) more efficiently, therefore yield per puff should increase.

L390-397: Furthermore, the TPM yield per puff should increase with power dissipated in the coil, based on first principles of heat and mass transfer. We did not observe this correlation between coil resistance and TPM yield. This suggests that some ENDS PCU may employ algorithms to overcome variation in coil resistance or actively control the power (either through current or voltage manipulation). Such algorithms could use a closed-loop control system which dynamically measures coil resistance and adjusts the power supplied to the coil in real time in order to keep the heating energy within a limit [50]. Thus, the temperature of the coil and heating chamber could be controlled in a cycle to keep aerosol emission steady.

Comment: Here could be another explanation. If power is already high, then the temperature of the heating will be limited by boiling point and further increase of the heating power should not affect the temperature as long as there is a liquid in contact with the coil. According to what you reported this type of ENDS does not have an overheating protection otherwise coil would not be burnt out.

L414-422: Concurrently, the e-liquid cools the coil as generated aerosol carries the heat away. This suggests that the presence of e-liquid around the coil contributes to limiting the coil temperature below its melting point, consistent with the observed steadiness in coil resistance values while there is e-liquid left in the pod (Figure 1). When insufficient e-liquid remains to fully submerge the coil and the wick, the heat generated by the coil remains in the coil and the wick causing the coil temperature to increase and thus the coil melts or breaks. A follow up experiment could be conducted to confirm this explanation by demonstrating the coil lifetime can be extended indefinitely by refilling the e-liquid reservoir, even though the pods studied here are intended by the manufacturer to be disposable. We observed the ENDS continued to operate during this pre-failure condition, when the coil was degraded (middle image) and loss of metal was observed.

Comment: Not clear what authors want to achieve. If to add liquid, then yes coil life will be extended but there is also so-called gunk accumulated on the coil with time plus coil itself would degrade. It is good that those ENDS are disposable. Or perhaps authors are wondering what would happen if users somehow will find a way to refill those disposable ENDS?

L426-432: Metal is likely ejected from the pod into the aerosol while the coil fails during the final puffing session and potentially much earlier. This is a potentially critical juncture, particularly in the instance where consumers misuse their product and refill the e-liquid in a pod which was nearly emptied on a prior use and whose coil had experienced degradation. When a compromised coil is subsequently heated, there may be increased risk of metal exposure even when the coil has not fully failed. Follow-up experiments to confirm the extension of coil lifetime are recommended, and, if successful, the presence of metal contaminants in the resulting aerosol may be investigated.

Comment: From the regulatory perspective product misuse is probably not a manufacturer responsibility. If user does manipulation (like refilling of disposable product) that is not recommended by manufacturer then it is not clear what FDA could do.

Author Response

Author response is in the file.

Reviewer 2 Report

The manuscript titled “Effects of Manufacturing Variation in Electronic Cigarette Coil Resistance and E-liquid Characteristics on Coil Lifetime and Aerosol Generation” (ijerph-1125266) describes measurements of coil resistance in Vuse e-cigarettes over the course of the lifetime of 15 pods as these are being “vaped”. The authors did not find correlations between initial coil resistance (Vuse pods come with the coil built into them) nor initial pod mass and coil lifetime and TPM yield. Even if such a correlation had been found, the sample size was small and this analysis was only carried out for one product, which uses an unusual coil setup (apparently not an actual “coil”). It remains unclear what impact such findings would have had beyond suggesting that pod manufacturing were inconsistent (which was not actually the case!). The manuscript then meanders into many speculative correlations, none of which are backed up by any data generated by the authors. Authors should consider actually measuring nicotine and nicotyrine concentrations if they wish to correlate these with their data, although it remains unclear to which extent nicotine is converted to nicotyrine. If this is in the percentage range in a low nicotine e-liquid, the nicotyrine would not likely have a large effect given its low concentration. In addition, simply judging e-liquid color “by eye” is not sufficient to describe pods, UV-VIS measurements would be needed and likely need to be correlated with GC/MS data on nicotine oxidation. However, that still raises the question how other compounds in e-liquids behave (oxidize?) and how this could contribute to color changes- the assumption that in such a complex mixture, only one oxidation occurs is likely too simplistic. Finally, some manuscripts on some of the topics already exist in literature, such as the finding that CPUs control e-cigarette power output (see for example for Juul: DOI 10.1038/s41598-020-64414-5 or 10.1136/tobaccocontrol-2018-054616) or e-liquid reactivity pre-heating (DOI 10.1093/ntr/nty192).

In conclusion, this reviewer’s recommendation is to reject the manuscript based on the outlined problems above.

Author Response

Author response is in the file.

Round 2

Reviewer 1 Report

Dear Authors,

Thank you for addressing my comments.

I think your manuscript can be published now.

Author Response

Thank you for reviewing our manuscript. Your suggestions made great improvements in the manuscript.  

Reviewer 2 Report

The revised manuscript titled “Effects of Manufacturing Variation in Electronic Cigarette Coil Resistance and E-liquid Characteristics on Coil Lifetime and Aerosol Generation” (ijerph-1125266) contains many fewer of the speculative statements that the authors had initially made, however, for this reviewer’s taste, there still remain speculative statements of hypotheses that authors may wish to prove or disprove experimentally, rather than speculating about:

  1. P8 L272: Authors speculate about the coil reaching the temperature of the boiling point of the liquid. Does the liquid have to boil to aerosolize? What about effects that cool the coil? This section seems to simplistic.
  2. P8 L278: Authors speculate that the “PCU did not exhibit an overheating circuit”. Authors could simply measure the voltage across the coil to understand what the power-control is actually doing in this scenario. It is known that Juul does exhibit over-heating protection, much like many of the “box-mod” e-cigarettes. Do authors have reason to believe the tested Vuse device does not feature such a function?
  3. Section 4.2: This author still believes it would be important to point out to readers that the “coil” in the examined device is not really a coil as one finds in most other e-cigarettes (e.g., a coiled metal wire of >5 turns), but rather a broader “metal bed” in s-shape. What implications does this different design have? Can results be translated to the more traditionally used actual “coil” setup?
  4. P9 l317: Authors seem to suggest that metal particles could be reaching the user upon coil failure: Could the authors please clarify this statement? Do they mean actual burning hot metal particles? Users would most definitely “feel” that, correct? If authors simply mean that users may be exposed to small quantities of metal, there is ample literature on the presence of metals in e-liquids and aerosol that would likely need to be mentioned in the text, then.
  5. P9 l321: Could authors please clarify: if a coil were degraded (e.g., high resistance), why do authors believe that it could be “heated” again at a later stage? In that case, would the e-cigarette simply not turn on? The released power would also be expected to be extremely low by the equation P=V^2/R. So it is unclear how users could be at “an increased risk of metal exposure”?

Author Response

Authors' responses are in the attached document.
